# Does CSR Action Provide Insurance-Like Protection to Tax-Avoiding Firms? Evidence from China

**Wei Li, Yuan Lu and Weining Li \***

School of Business Administration, South China University of Technology, Guangzhou 510641, China; lwroom@163.com (W.L.); lvyuan@scut.edu.cn (Y.L.)

**\*** Correspondence: adweinli@scut.edu.cn

**Abstract:** Based on a risk management perspective on corporate social responsibility (CSR), this study examines whether firms engaging in tax avoidance can benefit from CSR. We posit that CSR engagement can provide insurance-like protection for firm value by reducing the reputation risk of tax avoidance. Moreover, the extent to which CSR functions as insurance is largely dependent on a firm's communication strategy. In this study, a fixed-effect panel regression model is applied to examine the moderating effect of CSR engagement and greenwashing on the relationship between tax avoidance and firm value for listed Chinese firms. We find that a greenwashing strategy, i.e., a CSR communication strategy with aggressive symbolic actions and little to no substantive actions, generates negative capital and leads to a negative impact of tax avoidance on firm value. The findings are robust when considering deferred tax expenses and conducting a subgroup analysis. These findings advance our understanding of the relationship between tax avoidance, CSR and financial performance. They also help corporate executives select an effective CSR strategy for risk management purposes.

**Keywords:** tax avoidance; firm value; corporate social responsibility; risk management; greenwashing

## 1. Introduction

Tax research has shown that firms adopt various strategies to engage in tax avoidance, which appears to be highly effective in reducing costs and increasing investors' wealth [1]. However, recent studies have noted that tax avoidance is usually regarded as socially irresponsible and unethical by the public and the media [2–7]. Once tax avoidance behavior is publicly exposed, a firm may suffer penalties from tax authorities, negative news, consumers boycotts, stock price declines, and so on [6,8]. Therefore, tax avoidance increases risk, and ultimately leads to a negative evaluation by investors and a decrease in the firm value. In this way, tax avoidance can pose a significant risk to corporate reputation.

Given the reputation risk involved in tax avoidance, the literature has pointed out that tax-avoiding firms tend to adopt corporate social responsibility (CSR) practices to manage the risk [8–11]. CSR has been referred to as a firm's voluntary contribution to the development of a better society though various activities, such as philanthropy, environmental and/or charity projects, and the improvement of social welfare, beyond business operations [7,12]. In this vein, managers believe that CSR engagement can generate positive moral capital, which "can help temper stakeholders' negative judgments and their sanctions when adverse developments materialize" [13]. In other words, positive value of CSR against the risks caused by tax avoidance is usually referred to as "insurance-like protection" [14], which in turn makes the firm attractive to investors [15].

A majority of previous studies have focused on the relationship between tax avoidance and CSR interaction from the perspective of risk management which drives a firm's adoption of CSR. This is only half the story, though, since the consequences of adopting both tax avoidance and CSR remain unexplored. To what extent does CSR provide insurance-like protection for risks caused by tax

avoidance behavior and eventually enhance a firm's value? The question remains unanswered in prior research in this field.

This study has two purposes. The first concerns demonstrating the insurance-like mechanism of CSR engagement by assessing whether or not firms engaging in more CSR actions are more likely to benefit from tax avoidance. To answer this question, we hypothesized that the impact of tax avoidance on firm value depends on a firm's risk management capability. Financial research points out that investors will evaluate the risk and risk management level of the enterprise at the same time [16,17]. If the enterprise holds prior insurance, corporate executives' high-risk decisions are more likely to be supported by investors. Specifically, as CSR engagement can decrease the severity of reputation loss once firms are publicly inspected for being involved in tax avoidance [15], tax avoidance is more likely to be supported by investors, and firms are more likely to benefit from tax avoidance. In this study, therefore, we regard CSR engagement as an ex ante insurance mechanism and set out to demonstrate to what extent tax avoidance creates more value for firms engaging in CSR action.

The second aim of this study is to investigate the appropriate conditions for the CSR-generating insurance mechanism. Based on legitimacy theory, we argue that gaining moral legitimacy is necessary for CSR action to generate positive moral capital, which produces the insurance mechanism, but this does not mean that all CSR strategies gain moral legitimacy [18]. In this study, we take a CSR communication perspective by distinguishing between greenwashing and non-greenwashing strategies. Greenwashing refers to a communication strategy that overstates CSR practice and performance; that is, a firm demonstrates symbolic actions but little or no substantive actions [19,20]. Moral legitimacy is based on stakeholders' judgements about whether a firm is sincerely trying to improve social welfare [18,21]. However, a greenwashing strategy signals a firm's motivation to ingratiate themselves with stakeholders. That is, greenwashing reflects firms' intention to gain reputation by managing symbolic actions to camouflage a lack of effort to engage in true CSR activity. Therefore, greenwashing cannot generate moral legitimacy and, thus, cannot provide insurance-like protection for firms facing tax avoidance risks.

Two hypotheses are developed based upon previous studies, going on the assumption that CSR engagement and greenwashing strategy moderate the relationship between tax avoidance and firm value. We further tested these hypotheses by using a dataset of over 500 publicly listed Chinese firms from 2013 to 2016.

There are two main findings from the present study. First, CSR engagement has a positive moderating impact on the relationship between tax avoidance and firm value, implying that substantive CSR actions do have an insurance-like mechanism on firm value. Second, greenwashing has a negative moderating impact on the relationship between tax avoidance and firm value, implying that greenwashing generates negative moral capital and exacerbates the risks of tax avoidance.

These findings contribute to extant literature and practice. First, this study contributes to the research on the value of tax avoidance by introducing a risk management perspective on CSR. This study argues that tax avoidance is associated with a firm's reputation risk. A firm can benefit from tax avoidance by adopting CSR, as CSR actions act as a tool for managing reputation risk.

Second, this study contributes to the literature on the insurance-like mechanism of CSR by taking an ex ante perspective. This study argues that a firm can benefit from the insurance mechanism of CSR when no negative event has taken place, because CSR engagement signals a firm's risk management capability and, thus, reduces investors' evaluation of the reputation risk of tax avoidance.

Third, this study advances the research on the insurance-like mechanism of CSR from a CSR communication perspective. Previous studies have suggested that some conditions, such as the characteristics of stakeholders, firms, or events, could limit the insurance mechanism of CSR. This study adds a CSR communication dimension and argues that managers should be careful about choosing CSR communication strategies. We introduce the concept of greenwashing, arguing that it may lead to stakeholders' negative evaluation of a firm's CSR motive. Thus, it will limit the insurance-like mechanism of CSR on tax avoidance.

Forth, this paper focused on the Chinese context. Nowadays, in China, it is increasingly a common view that, although rapid economic growth is important, firms should assume more social and environmental responsibilities. In 2006, the State Council introduced a CSR reporting system to Chinese state-owned enterprises. Since 2012, all central enterprises have been required to report CSR activities and performance. As a result, Chinese firms have adopted CSR as a strategy to gain reputation and legitimacy. Although previous studies commonly focused on the direct impact of CSR on financial performance, the present study approached CSR from a risk management perspective to investigate whether CSR could protect the firm value. Moreover, previous studies examined the insurance-like mechanism of CSR in the context of developed countries, such as the United States, but few studies have examined the insurance mechanism of CSR in transition economies like China. Our study will fill the gap by focusing on CRS in the Chinese context.

The remainder of this paper will be structured as follows: Section 2 offers a theoretical overview and describes the hypothesis development; Section 3 outlines the methodology of the paper; Section 4 highlights the research findings; Section 5 conducts robustness tests; Section 6 contains a discussion of the research findings and management suggestions; and the final section is devoted to the conclusions.

## 2. Theoretical Background and Hypotheses

In this section, we provide a quick overview of the literature on the reputation risks of tax avoidance and the insurance mechanism of CSR. Then, we discuss how CSR actions mitigate the risks caused by tax avoidance.

### 2.1. The Reputation Risks of Tax Avoidance

Corporate executives may be motivated to take a variety of measures to avoid paying tax because tax avoidance can save costs, increase profits, and enhance the value of enterprises [22,23]. However, the evidence indicates that there is wide variation in the level of tax avoidance across firms; some firms engage in it aggressively, while others appear to shun it. The question of why so many firms do not take opportunities to engage in tax avoidance has been raised as the "under-sheltering puzzle" [24,25].

To solve this puzzle, the accounting literature often posits reputation risk as an important factor limiting tax avoidance activities. Scholars suggest that a firm's tax avoidance is considered as both illegal and unethical by their stakeholders [2,4]. A firm engaging in tax avoidance may not only be found out by the tax authorities and forced to pay additional taxes and penalties, but also be labeled as a "poor corporate citizen" by the public [2]. The situation becomes worse if the news of tax avoidance is disclosed or published by the media: the firm's reputation will be damaged, which, in turn, may result in a chain of actions from stakeholders, such as customer boycotts [26] and suppliers' shift to competitors [27].

In this way, tax avoidance will have a negative impact on investors' willingness to continue investing in or supporting the firm. Hanlon and Slemrod [3], for instance, have examined the stock price responses of firms accused of engaging in tax avoidance and found that firms suffered stock price declines following the public revelation of tax avoidance behavior.

Recently, given the reputation risk of tax avoidance, scholars developed a risk management perspective on the relationship between tax avoidance and CSR. This perspective treats CSR as a risk management tool that a firm uses to enhance its reputation, which, in turn, lessens the expected reputation loss associated with tax avoidance practices [3].

In line with this perspective, this study examines the moderating role of CSR on the effect of tax avoidance on firm value, where CSR can be viewed as an insurance mechanism for a firm's reputation.

### 2.2. Risk Management Perspective and the Insurance Mechanism of CSR Engagement

Previous studies have extensively studied the relationship between CSR engagement and financial performance and have often assumed that the value of CSR engagement lies in the enhancing corporate reputation [28], generating greater consumer support [29], deepening employee commitment [30],

promoting legitimacy [16], and developing better governmental relationships [29]. However, there has been little research concerning the risk management role performed by CSR and few related empirical studies [31].

According to the risk management perspective, CSR creates value by protecting corporate financial performance rather than promoting it [32]. Recently, empirical studies have increasingly found a negative relationship between firm risks (e.g., stock price crash risk and firm default risk) and CSR [33,34].

Previous research has provided two main explanations for the risk-management mechanism of CSR. First, CSR can directly reduce the firm's risk exposure by avoiding harm to stakeholders [35]. For example, the adoption of a CSR management system can help firms develop their capabilities in emergency preparedness and responses, while environmental pollution monitoring and creating employees' awareness and competence [35].

Second, CSR can indirectly reduce firm risk by managing relationships with their stakeholders. On the one hand, CSR facilitates closer firm–stakeholder relationships, which helps firms access more information [36,37], makes firms more sensitive to changes or threats [38], and enables firms more capability to anticipate and prevent foreseeable risks. On the other hand, CSR can build positive moral capital among stakeholders, providing a buffer for a firm's relationship-based intangible assets against loss by mitigating negative assessments [15]. For example, in the event of a crisis, CSR engagement can affect consumers' attributions of blame and can consequently mitigate negative brand evaluations [39] or reduce reputation loss [40].

This calls attention to the insurance mechanism of CSR for protecting investors' wealth from negative events. A handful of studies have investigated how CSR affects investors' wealth after a negative event [41–45]. For instance, Godfrey and colleagues [16] studied 178 negative events between 1993 and 2003 and found that the negative impact of such events is smaller for firms participating in CSR. Shui and Yang [45] have investigated 1745 negative events affecting firms in 2001–2008 and found that engagement can mitigate the decline of firms' stock and bond price during occurrences of negative events. These studies used samples of firms that had experienced negative events and demonstrated the ex post insurance mechanism of CSR.

Recently, scholars have noted that CSR could serve as an ex ante insurance in promoting firm value even if no negative event has happened [11,39]. Scholars in the financial literature argued that investors evaluate both firms' perceived risk exposure and the mechanisms that reduce the risk [15,16]. Investors' positive evaluation of firm value is associated with firms' low risk exposure or high-level risk management capability [13,46]. Given the insurance mechanism of CSR on the potential reputation loss caused by tax avoidance, CSR engagement can mitigate investors' negative evaluation of tax avoidance. We thus propose the following hypothesis:

**Hypothesis 1.** *CSR engagement positively moderates the impact of tax avoidance on firm value.*

### 2.3. Greenwashing Strategy

Relevant to the proceeding discussion, in order to make CSR generate positive moral capital, the firm must gain moral legitimacy to meet certain basic conditions [13,15]. Moral legitimacy reflects stakeholders' normative approval of a firm's CSR actions [18,21]. If CSR actions can be evaluated as genuine manifestation of firms' intention in improve overall social welfare, positive moral capital will be generated [16]. By contrast, if CSR actions are evaluated as an attempt to pursue self-interest by ingratiation, the firm may less likely to gain positive moral evaluation and even generate negative moral capital [15]. Previous studies implied that the extent to which the firm obtains moral capital is largely contingent on its organizational legitimacy, which is a result of CSR strategy.

According to the management literature, CSR is an organizational legitimacy action aimed at winning the support of institutional constituents, including government, investors, shareholders,

and other stakeholders, such as the public and consumers [47]. To gain legitimacy, firms may adopt various types of strategies to communicate with stakeholders. Recently, academics have noted that cases of firms adopting a greenwashing strategy deserve attention [19]. The concept of greenwashing was initially applied to the narrow field of environmental issues, but many researchers have related greenwashing to social and economic issues and assigned it a broader meaning [48]. Greenwashing is defined as a strategy where firms aggressively use a symbolic measure to communicate about their CSR performance with little to no substantive actions to back it up [19]. A substantive CSR action is costly as it involves resource allocation and makes an actual change in operations and organizational structure. By contrast, a symbolic action is merely ceremonial, so that the firm can be seen to comply with rules, policies, and the public's expectations [20].

A firm can obtain benefits from conducting a greenwashing strategy, such as saving costs, and moderating the conflict between the pursuit of maximum profit and secondary stakeholders' pressure [20]. However, there exist costs for taking a greenwashing strategy. If a greenwashing firm was exposed, the firm would suffer from a decline of financial performance [21,48].

Researchers have argued that a greenwashing strategy will lead to confusion and skepticism among stakeholders [49], as greenwashing firms may be viewed as untrustworthy, manipulative, and opportunistic [50]. There is empirical evidence demonstrating stakeholders' negative response to greenwashing. For instance, Nyilasy, Gangadharbatla, and Paladino [51] found that a greenwashing communication strategy could negatively affect firms' brand value. Du [48] found that firms' greenwashing strategy was not able to trick investors and they would be penalized by the financial market.

According to Godfery [16], negative moral capital arises when the act or actors receives a negative evaluation from the target stakeholders. As greenwashing cannot pass the moral legitimacy test and will generate negative moral capital, it will exacerbate the reputation risk caused by tax avoidance.

Based on the above analysis, greenwashing is a signal of an ineffective insurance mechanism and high risks of tax avoidance to investors. Investors will evaluate tax avoidance negatively when a firm has adopted a greenwashing strategy. We thus propose the following hypothesis:

**Hypothesis 2.** *Greenwashing negatively moderates the impact of tax avoidance on firm value. In other words, the negative effect of tax avoidance on firm value will be greater for greenwashing firms.*

## 3. Research Methodology

### 3.1. Model

According to Morgan [52], if the p value of the Hausman test is less than 5%, the fixed-effect model should be used to analyze the panel data. Otherwise, the random-effect model should be applied. The Hausman test ($p < 5\%$) is appropriate for the fixed-effects model. Thus, a fixed-effect panel regression model was used to test the hypotheses of this study. The regression model is defined as follows:

$$TQ_{it+1} = \beta_0 + \beta_1 TA_{it} + \beta_2 CSR_{it} + \beta_3 GW_{it} + \beta_4 TA_{it} \times CSR_{it} + \beta_5 TA_{it} \times GW_{it} + \beta_6 X_{it} + indus_i + year_t + \varepsilon_{it} \quad (1)$$

$TQ_{it+1}$ is a proxy of firm value measured as Tobin's Q; $TA_{it}$ is a proxy of tax avoidance; $CSR_{it}$ is a proxy of CSR engagement; $GW_{it}$ is a proxy of greenwashing; and $X_{it}$ is a vector of control variables. $indus_i$ and $year_t$ are industry and time dummies. $\varepsilon_{it}$ is an error term.

The STATA software package was used to run the regression.

### 3.2. Sample and Data Resource

Our sample came from 2132 observations of listed firms on Shenzhen and Shanghai stock exchanges between 2013 and 2016. We selected our sample based on the following criteria: (1) observations with

negative net assets or owners' equity dropping; (2) we exclude observations that have been listed for less than one year; (3) we also exclude those whose data were incomplete or missing. Finally, we arrived at a sample of 1993 firm-year observations.

Data sources are reported below. The data on tax avoidance, firm value, enterprise characteristics, and financial information of listed companies were derived from the China Stock Market and Accounting Research (CSMAR) database. CSMAR is a widely used database for information on financial position of Chinese firms. The data on CSR engagement were collected from CSR rating agency Rankins CSR Ratings (RKS) database. Data on CSR communication strategies were hand-collected from firms' CSR/sustainability reports and annual reports by employing content analysis and conducting scoring procedures.

### 3.3. Dependent Variable

#### Firm Value

According to extensive literature in corporate finance, including Demsetz and Lehn [53], Mehran [54], and Morck, Shleifer and Vishny [55], Tobin's Q has been popularly used to measure firm value. In the present study, we decided to use Tobin's Q (TQ) as a proxy for firm value, because Tobin's Q reflects the potential total value of a firm in the investment market [56–58] and enabled us to capture whether or not investors value a firm's tax avoidance and CSR actions positively.

According to previous studies, TQ is defined as

$$TQ = \frac{Market\ value\ of\ equity + Short\ Term\ Debt + Long\ Term\ Debt}{Total\ Assets} \tag{2}$$

To decrease the weight of extreme outliers, TQ is winsorized at the 1st and 99th percentiles.

### 3.4. Independent Variable

#### Tax Avoidance

The previous literature proposed a number of indicators to measure tax avoidance [59]. We constructed a variable, *TA*, to measure tax avoidance. *TA* is the difference between the statutory tax rate and the *Cash ETR* for measuring tax avoidance. The *Statutory Tax Rate* is the tax rate specified by law. *Cash ETR* is estimated as the ratio between cash taxes paid per and the book value of pretax income [3,10]. See the formula below:

$$TA = Statutory\ Tax\ Rate - Cash\ ETR = Statutory\ Tax\ Rate - \frac{Cash\ Tax\ Paid}{Pretax\ Income} \tag{3}$$

According to this formula, the higher the difference between the *Statutory Tax Rate* and *Cash ETR*, the more likely it is that a firm has engaged in tax avoidance. Furthermore, the present study was arranged in China, where the government sets different statutory tax rates for different industries. For example, high-technology firms enjoy lower statutory tax rates than do firms in traditional industries. Therefore, the introduction of a statutory tax rate to measure tax avoidance enables us to capture the degree to which a firm may have violated the tax law and the risk involving tax authorities' public scrutiny. Therefore, the above formula is assumed to be reasonable for measuring tax avoidance.

To decrease the weight of extreme outliers, *TA* is winsorized at the 1st and 99th percentiles.

### 3.5. Moderators

#### 3.5.1. CSR Engagement Measure

We used the RKS rating database to measure firms' CSR engagement. RKS is a third-party CSR rating organization and is one of the largest CSR databases in China. The RKS rating standard is based on the GRI (Global Reporting Initiative) guidelines, which assesses firms' CSR performance on several

dimensions, including CSR plans, corporate governance, safety, environment, employee, customer, philanthropy and community. RKS rating has been adopted in many China-specific studies [60,61].

3.5.2. CSR Communication Strategy: Greenwashing vs. Non-Greenwashing

As discussed in preceding sections, the CSR communication strategy concerns the configuration of symbolic and substantive actions. According to Schons and Steinmeir [62], the intersection of symbolic and substantive CSR actions generates four configuration strategies: mere talk (greenwashing), mere walk, walk the talk, and neither walk nor talk. In this study, we are interested in firm's window dressing motives, as signaled by the CSR communication strategy; thus, we measured the degree of a firm's involvement in greenwashing. According to Walker and Wan [21], we classified firms that demonstrate symbolic actions but little to no substantive actions as greenwashing firms. Our assessment of greenwashing used the following procedure.

First, we built a substantive actions index and symbolic actions index. By employing content analysis, we identified the substantive and symbolic actions mentioned in firms' CSR/sustainability reports and annual reports. Following the idea of Testa et al. [63], we measured the substantive actions index, rating firms' CSR key performance indicators and main costs; we measured the symbolic actions index by rating firms' ceremonial arrangements that are not necessarily implemented. Details of their definition and rating methods are shown in Table 1. To ensure that the substantive actions index and the symbolic actions index can be compared, both of them are standardized to have a standard deviation of 1 and a mean of 0 [64].

**Table 1.** Definition of substantive and symbolic corporate social responsibility (CSR) actions.

| Index | Description and Scoring Method |
|---|---|
| Substantive actions index | Energy and resource efficiency (To what extent does the firm report on its engagement to improve energy and resource efficiency?) (0–10); |
| | Emissions and waste reduction (To what extent does the firm report on their effort to recycle resources, treat or reduce emissions and wastes?) (0–10); |
| | Green product development (To what extent does the firm report on environmentally-friendly products or services?) (0–10); |
| | Employee career development (To what extent does the firm report on employee training and self-improvement?) (0–10); |
| | Employee rights (To what extent does the firm report on employee compensation, welfare, occupation health, and safety?) (0–10); |
| | Quality of products and services (Has the firm passed an international quality management system certification?) (0–1); |
| | Philanthropic donation (Does the firm report on the amount of philanthropic donations?) (0–1); |
| | Participation in science and technology development (To what extent does the firm report on participating in universities' and research institutes' projects?) (0–10). |
| Symbolic actions index | CSR vision, strategy and value claims (To what extent does the firm report on its vision, strategy and values?) (0–10); |
| | Governance structure and management systems (To what extent does the firm describe its CSR management teams and management activities?) (0–10); |
| | Communication with stakeholders (To what extent does the firm explain how it manages the relationship with stakeholders?) (0–10); |
| | GRI report guidelines (Is the firm's CSR report published in accordance with the GRI guidelines?) (0–1); |
| | External audit (Has the CSR report been reviewed by an external auditor?) (0–1); |
| | Global compact (Has the firm joined the Global Compact network?) (0–1). |

Second, following Walker and Wan [21], we built a variable greenwashing index (GW), which is measured as the difference between the standardized symbolic actions index and the standardized substantive actions index. It captures the discrepancy between firm-level symbolic and substantive actions. The higher the GW, the more likely it is that a firm adopts a greenwashing strategy; the lower the GW, the more likely it is that a firm adopts a non-greenwashing strategy.

### 3.6. Controls

The present study selected the following control variables. Return on assets (ROA) is controlled because return on assets usually reflects a firm's efficiency of operations, and investors tend to hold shares in firms with a higher return on assets. The asset–liability ratio (LEVER) is controlled because it refers to a firm's financial risk, which affects investors' evaluation of the listed firms. SIZE, measured by the logarithmic value of total assets, is controlled because larger firms tend to have stronger market competitiveness and are more likely to resist financial risks. TOP1 is the shareholding ratio of the largest shareholder of the listed firm, which is controlled because the corporate governance structure may affect the firm value. MK (the stock market) is a dummy variable of listing location.

Two stock markets in China are concerned, which are the Shanghai stock exchange and the Shenzhen stock exchange. To control the influence of institutional environment factors on firm value, we measured the firms listed in the Shanghai stock exchange as 1, and the firms in the Shenzhen stock exchange as 0. LOCAL (location of the firm) is a dummy variable classifying developed regions and underdeveloped region. China is a country with an imbalance of regional economy development: firms located in developed regions are likely to take advantage of more resources, a better market system, and more opportunities. Thus, we controlled the effect of location on firm value. LOCAL equals 1 for firms headquartered in regions where the per capita GDP has been in the top 10 for five years and 0 otherwise. SOE (state owned enterprise) equals 1 if the firm is ultimately owned by the government. In China, government bodies are very powerful and control many critical resources, and state-owned enterprises have priority access to strategic resources controlled by the government. Thus, state-owned enterprises are more likely to have a higher firm value.

We also include year and industry dummy variables (fixed effects) in our regressions. According to the "Guidelines on Industry Classification of Listed Firms" issued by the China Securities Regulatory Commission (CSRC) in 2001, we defined 11 industries in our sample.

## 4. Research Methodology

### 4.1. Descriptive Statistics

Table 2 shows the descriptive statistics of the main variables. (1) CSR ranges from 12.90 to 67.33, indicating that the sample enterprises have great heterogeneity in CSR engagement. GW ranges from −1.89 to 2.03, capturing the heterogeneity of firms' communication CSR strategies adoption based on their symbolic and substantive actions. Tax avoidance ranges from −0.52 to 0.62, including both positive and negative values, indicating the existence of tax avoidance. ROA ranges from −0.69 to 7.45, which indicates that the sample includes not only enterprises with better performance, but also the enterprises whose performance is declining. Firm SIZE (total assets) ranges from 0.22 to 24,100 billion RMB, which indicates that the sample includes both super-large enterprises and small-scale enterprises.

Table 3 reports the Pearson correlations between the main variables. The correlation coefficients are lower than 0.5, which is lower than the threshold of judging multicollinearity. Thus, the subsequent results will not be affected by the multicollinearity problem.



**Table 2.** Descriptive statistics of variables.

| Variable | N | Mean | Median | SD | Min | Max |
|---|---|---|---|---|---|---|
| Tobin's Q (TQ) | 1993 | 1.30 | 0.96 | 1.19 | 0.05 | 6.75 |
| TA | 1993 | 0.05 | 0.04 | 0.14 | −0.52 | 0.62 |
| CSR | 1993 | 32.75 | 30.58 | 10.03 | 12.90 | 67.33 |
| Greenwashing index (GW) | 1993 | −0.01 | 0.01 | 0.56 | −1.89 | 2.03 |
| Return on assets (ROA) | 1993 | 0.03 | 0.02 | 0.17 | −0.69 | 7.45 |
| Asset–liability ratio (LEVER) | 1993 | 0.55 | 0.57 | 0.21 | 0.02 | 1.34 |
| Total assets (SIZE) | 1993 | 280 | 17 | 1700 | 0.22 | 24,100 |
| TOP1 | 1993 | 39.15 | 39.36 | 16.38 | 3.39 | 89.41 |
| LOCAL | 1993 | 0.52 | 0 | 0.49 | 0 | 1 |
| MK | 1993 | 0.76 | 0 | 0.42 | 0 | 1 |
| SOE | 1993 | 0.31 | 0 | 0.46 | 0 | 1 |

**Table 3.** Correlation matrix of main variables.

| | TA | CSR | GW | ROA | LEVER | SIZE | TOP1 | LOCAL | MK | SOE |
|---|---|---|---|---|---|---|---|---|---|---|
| TA | 1 | | | | | | | | | |
| CSR | −0.0254 | 1 | | | | | | | | |
| GW | 0.0265 | −0.4321 *** | 1 | | | | | | | |
| ROA | −0.0537 ** | −0.0117 | 0.0574 *** | 1 | | | | | | |
| LEVER | 0.0898 *** | 0.2206 *** | −0.1055 ** | −0.1352 *** | 1 | | | | | |
| SIZE | −0.0114 | 0.4350 *** | −0.2846 *** | −0.0605 *** | 0.4582 *** | 1 | | | | |
| TOP1 | −0.0126 | 0.1392 *** | −0.0962 *** | 0.0192 | −0.0251 | 0.1393 *** | 1 | | | |
| LOCAL | −0.0670 *** | 0.1711 *** | −0.0932 | 0.0200 | 0.0232 | 0.1191 *** | 0.0542 | 1 | | |
| MK | 0.0320 | 0.0002 | −0.0722 ** | 0.0085 | 0.0176 | 0.0119 | 0.0739 *** | 0.1139 *** | 1 | |
| SOE | 0.0898 *** | 0.2206 *** | 0.1711 * | 0.1667 * | −0.0932 *** | 0.0200 | 0.0232 | 0.0542 | 0.2260 *** | 1 |

Note: All *p* values reported are at two-tailed significance; * $p < 0.10$, ** $p < 0.05$, and *** $p < 0.01$.

## 4.2. Direct Effect of Tax Avoidance on Firm Value

This study assumes that investors perceive corporate tax avoidance practice as a two-edged sword: it involves benefits, but also reputation risk. Model (1) in Table 4 tests this assumption. In model (1), the coefficient of TA is negative and significant ($p < 0.5$). The result shows that the negative effect of tax avoidance on investors' wealth exceeds its positive effect, consistent with the notion that tax avoidance is associated with reputation risk and a decline in firm value [3].

## 4.3. Interaction Effects between CSR Actions and Tax Avoidance on Firm Value

Hypothesis 1 assumes that CSR engagement positively moderates the impact of tax avoidance on firm value. Model (3) in Table 4 tests this hypothesis. In model (3), the coefficient of TA × CSR is positive and significant ($p < 0.1$). Following Aiken and West [65], we plot the interaction effect of CSR engagement and tax avoidance. Figure 1 depicts a positive relationship between tax avoidance and firm value for firms with more CSR engagement, while there is a negative relationship between tax avoidance and firm value for firms with less CSR engagement. This implies that the CSR engagement is valued by investors as effective insurance. With the ex ante insurance mechanism of CSR, a firm's involvement in tax avoidance is likely to be assessed positively. This provides support for Hypothesis 1.



**Table 4.** The impact of tax avoidance on firm value and the moderating effect of CSR engagement and greenwashing.

|  | **Variables** | **(1)** | **(2)** | **(3)** | **(4)** | **(5)** | **(6)** |
|---|---|---|---|---|---|---|---|
| Independent variables | TA | −0.139 ** (−2.34) | −0.138 ** (−2.33) | −0.132 ** (−2.17) | −0.897 *** (−2.94) | −0.081 (−1.33) | −1.603 *** (−4.88) |
| | CSR | | −0.017 *** (−5.95) | | 0.014 *** (0.341) | | −0.009 * (−2.72) |
| | GW | | | 0.058 *** (6.95) | | 0.159 *** (2.80) | 0.015 (1.40) |
| Moderating effect | TA × CSR | | | | 0.024 ** (2.53) | | 0.049 *** (4.70) |
| | TA × GW | | | | | −0.509 *** (−3.95) | −0.838 *** (−5.37) |
| Controls | ROA | 0.562 *** (8.92) | 0.566 *** (8.88) | 0.560 *** (8.86) | 0.566 *** (8.88) | 0.579 *** (9.10) | 0.548 *** (8.67) |
| | LEVER | −1.790 *** (−10.29) | −1.797 *** (−10.35) | −1.817 *** (−10.64) | −1.797 *** (−10.35) | −1.810 *** (−10.44) | −1.794 *** (−10.32) |
| | SIZE | −0.349 *** (−15.66) | −0.347 *** (−15.53) | −0.328 *** (−15.56) | −0.347 *** (−15.53) | −0.328 *** (−15.59) | −0.354 *** (−15.70) |
| | TOP1 | 0.004 *** (3.23) | 0.004 *** (3.20) | 0.004 *** (3.34) | 0.004 *** (3.20) | 0.004 *** (3.36) | 0.004 *** (3.10) |
| | LOCAL | 0.125 *** (3.17) | 0.124 *** (3.13) | 0.140 *** (3.63) | 0.124 *** (3.13) | 0.141 *** (3.65) | 0.117 *** (2.95) |
| | MK | −0.010 (−0.21) | −0.005 (−0.11) | −0.007 (−0.15) | −0.005 (−0.11) | −0.004 (−0.08) | 0.015 (0.33) |
| | SOE | 0.014 *** (3.42) | 0.013 *** (3.22) | 0.014 *** (3.42) | 0.013 *** (3.01) | 0.014 *** (3.31) | 0.013 *** (3.12) |
| Fixed effect | Year | Yes | Yes | Yes | Yes | Yes | Yes |
| | Industry | Yes | Yes | Yes | Yes | Yes | Yes |
| | Constant | 11.059 *** (23.38) | 11.198 *** (22.40) | 11.438 *** (23.10) | 11.214 *** (22.42) | 11.156 *** (22.34) | 11.261 *** (21.32) |
| | $R^2$ | 0.5123 | 0.5133 | 0.5121 | 0.5128 | 0.5135 | 0.5175 |
| | Adj $R^2$ | 0.5071 | 0.5078 | 0.5069 | 0.5133 | 0.5081 | 0.5116 |
| | $F$ | 155.02 | 137.93 | 154.24 | 146.29 | 136.89 | 114.90 |

Note: All $p$ values reported are at two-tailed significance; * $p < 0.10$, ** $p < 0.05$, *** and $p < 0.01$. N = 1993.

**Figure 1.** Interaction between CSR engagement and tax avoidance.

Hypothesis 2 assumes that greenwashing negatively moderates the impact of tax avoidance on firm value. Model (4) in Table 4 tests this hypothesis. In model (4), the coefficient of TA × GW is negative and significant ($p < 0.05$). Figure 2 shows that there is a positive relationship between tax avoidance and firm value for non-greenwashing firms, while there is a negative relationship between tax avoidance and firm value for greenwashing firms. Hypothesis 2 is therefore supported.

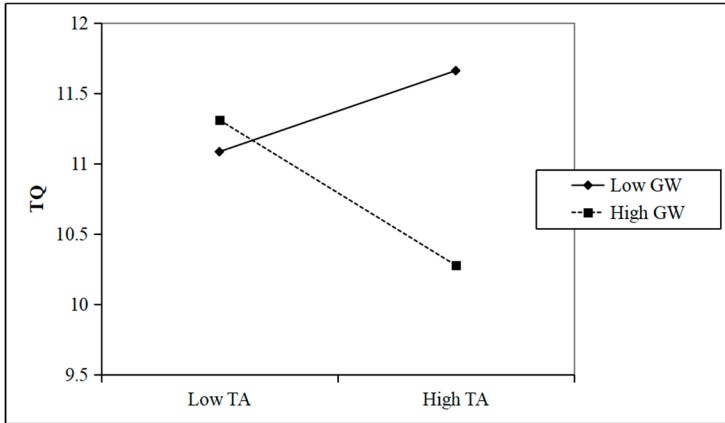

**Figure 2.** Interaction between greenwashing and tax avoidance.

Model (5) in Table 4 contains both the effect of CSR engagement and greenwashing, and the result is consistent with model (3) and (4). There is no significant change in the significance of the coefficient.

## 5. Robustness Tests

### 5.1. An Alternative Measure of Tax Avoidance

Before the implementation of the new accounting standards in 2006, there were few deferred income tax charges for enterprises in China. However, firms began to use a balance sheet approach to calculate corporate income tax and recognize deferred tax after 2006. This has led to a significant increase in deferred tax expense for enterprises since 2007. Tax avoidance activities may occur in the form of deferred tax in the future, thus increasing the impact of deferred tax expense on the effective tax rate. After dropping those observations where deferred tax expense data are missing, we had a sample of 1969 observations.

TA values used in the preceding text have not taken deferred tax expenses into account, which may cause errors in the measurement. Accordingly, we used variable TAD (tax avoidance with deffered tax paid) as an alternative measure of tax avoidance to re-test the hypotheses.

The formula of tax avoidance considering deferred tax is:

$$TAD = \ Statutory\ Tax\ Rate - \frac{Cash\ Tax\ Paid - Deffered\ Tax\ Paid}{Pretax\ Income} \tag{4}$$

The results of the robustness test are shown in Table 5, and there is no significant change in the significance of the model coefficient and the positive and negative properties, indicating that the results obtained above are robust.

**Table 5.** Robustness test by using an alternative measure of tax avoidance.

| | Variables | (1) | (2) | (3) | (4) | (5) | (6) |
|---|---|---|---|---|---|---|---|
| Independent variable | TAD | −0.128 * (−1.80) | −0.134 (−1.57) | −0.128 (−1.51) | −0.313 (−1.10) | −0.406 ** (−2.31) | −0.490 (−1.60) |
| | CSR | | 0.021 *** (4.52) | | 0.020 *** (4.23) | | 0.021 *** (4.29) |
| | GW | | | 0.009 (1.33) | | 0.012 * (1.81) | 0.015 ** (2.20) |
| Moderate effect | TAD × CSR | | | | 0.032 ** (2.73) | | 0.007 ** (2.38) |
| | TAD × GW | | | | | −0.074 ** (−2.30) | −0.069 ** (−2.15) |
| Controls | ROA | | 0.540 *** (8.52) | 0.559 *** (8.85) | 0.538 *** (8.54) | 0.575 *** (9.06) | 0.545 *** (8.58) |
| | LEVER | | −1.745 *** (−10.13) | −1.781 *** (−10.28) | −1.748 *** (−10.13) | −1.762 *** (−10.20) | −1.742 *** (−10.09) |
| | SIZE | | −0.364 *** (−16.01) | −0.329 *** (−15.54) | −0.363 *** (−15.95) | −0.329 *** (−15.59) | −0.356 *** (−15.84) |
| | TOP1 | | 0.003 *** (2.94) | 0.004 *** (3.21) | 0.003 *** (2.93) | 0.004 *** (3.23) | 0.004 *** (2.96) |
| | LOCAL | | 0.112 *** (2.82) | 0.140 *** (3.61) | 0.112 *** (2.79) | 0.141 *** (3.66) | 0.118 *** (2.97) |
| | MK | | −0.001 (−0.01) | −0.012 (−0.26) | 0.001 (0.02) | −0.006 (−0.883) | 0.012 (0.25) |
| | SOE | | 0.014 *** (3.43) | 0.014 *** (3.42) | 0.013 *** (3.22) | 0.014 *** (3.31) | 0.013 *** (3.12) |
| Fixed effect | Year | | Yes | Yes | Yes | Yes | Yes |
| | Industry | | Yes | Yes | Yes | Yes | Yes |
| | Constant | | 11.262 *** (22.10) | 11.228 *** (22.23) | 11.123 *** (22.17) | 11.190 *** (22.24) | 11.197 *** (22.41) |
| | $R^2$ | | 0.5088 | 0.5052 | 0.5089 | 0.5076 | 0.5118 |
| | Adj $R^2$ | | 0.5035 | 0.4999 | 0.5033 | 0.5020 | 0.5157 |
| | $F$ | | 152.14 | 150.53 | 135.59 | 133.91 | 112.34 |

Note: All $p$ values reported are at two-tailed significance; * $p < 0.10$, ** $p < 0.05$, *** and $p < 0.01$. N = 1969.

### 5.2. Subgroup Analysis as an Alternative Method to Test Moderating Effect

We also conducted a subgroup analysis as an alternative method to examine the moderating effect of CSR engagement and greenwashing. According to Testa et al. [63], observations in the top 75th percentile of CSR belong to the high-level group, and observations in the bottom 25th percentile of CSR belong to the low-level group. Observations of the top 75th percentile of GW belong to greenwashing group. Otherwise, they belong to the non-greenwashing group. The results show that TA has a significant positive effect on TQ ($p < 0.05$) for firms with high-level CSR engagement, while TA has a significant negative effect on TQ ($p < 0.10$) for firms with low-level CSR engagement (see Table 6). Moreover, TA has an non-significant effect on TQ for non-greenwashing firms, while TA has a significant negative effect on TQ ($p < 0.01$) for greenwashing firms (see Table 6). The results of subgroup analyses are basically consistent with the results above.

**Table 6.** Robustness test by conducting subgroup analyses.

| | Variables | Firms with High-Level CSR Engagement | Firms with Low-Level CSR Engagement | Greenwashing Firms | Non-Greenwashing Firms |
|---|---|---|---|---|---|
| Independent variable | TA | 0.404 ** (1.97) | −0.301 * (−1.87) | −1.664 *** (−4.34) | 0.015 (0.12) |
| Controls | ROA | 4.552 *** (5.28) | 0.278 *** (8.88) | 0.415 ** (2.01) | 2.361 *** (2.91) |
| | LEVER | −1.384 *** (−5.25) | −1.418 *** (−4.45) | −1.088 ** (−2.27) | −1.611 *** (−5.87) |
| | SIZE | −0.201 *** (−8.79) | −0.701 *** (−12.55) | −0.664 *** (−8.58) | −0.283 *** (−8.56) |
| | TOP1 | 0.002 (0.95) | 0.006 * (1.71) | 0.011 ** (2.33) | 0.003 (1.43) |
| | LOCAL | 0.026 (0.40) | 0.029 (0.25) | −0.120 (−0.71) | 0.192 *** (2.66) |
| | MK | 0.133 * (1.89) | −0.195 (−1.37) | −0.025 (−0.14) | −0.115 (−1.14) |
| | SOE | 0.012 *** (3.13) | 0.013 *** (3.22) | 0.015 *** (3.17) | 0.011 *** (3.05) |
| Fixed effect | Year | Yes | Yes | Yes | Yes |
| | Industry | Yes | Yes | Yes | Yes |
| | Constant | 11.059 *** (11.80) | 18.137 *** (15.18) | 17.437 *** (10.80) | 9.584 *** (9.27) |
| | $R^2$ | 0.5123 | 0.5910 | 0.5315 | 0.5910 |
| | Adj $R^2$ | 0.5071 | 0.5709 | 0.5056 | 0.5709 |
| | $F$ | 29.41 | 27.47 | 20.48 | 20.75 |
| | $N$ | 428 | 522 | 433 | 481 |

Note: All *p* values reported are at two-tailed significance; * $p < 0.10$, ** $p < 0.05$, and *** $p < 0.01$.

### *5.3. Examine the Hypotheses with the Propensity Score Matching (PSM) Method*

The present study concerns the endogeneity problem caused by sample selection bias [66]. Specifically, firms with lower firm value may be more inclined to engage in tax avoidance. Consequently, the observed firm's value decline may be driven by reasons other than tax avoidance. Therefore, we apply the propensity score matching (PSM) approach to estimate whether tax avoidance really impacts firm value. PSM enables us to find matched samples with characteristics similar to those of the treatment group from a control group. Furthermore, it helps us to reduce the sample selection bias [67]. The procedure of PSM involves the following steps:

First, we identified the tax-avoidance group as the treatment group and non-tax-avoidance group as control group. The tax-avoidance group includes observations of the top 75th percentile of the TA variable, whereas the non-tax-avoidance group includes other observations. Our final dataset contains 522 observations in the tax-avoidance group and 1471 observations in the non-tax-avoidance group.

Second, we calculated propensity scores (PSs), which indicate the propensity of tax avoidance engagement. To estimate PSs, we adopted a Probit regression model [68], and PSs are the predicted values of the model. The indicator variable equals 1 for firms in the tax-avoidance group and 0 otherwise. There are seven independent variables that can affect tax avoidance: ROA; total assets (SIZE); total debt over total assets (LEV); net cash from operating activities (CF); plant, property, and equipment over total assets (PE); intangible asset over total assets (IA); and sales growth rate (SALE) [7,26].

Third, based on PS values, Nearest-Neighbor Matching (NNM), Radius Matching (RM), and Kernel Matching (KM) method are applied in this study. After matching, following Becker and Ichino [69], we calculated the average effect of treatment on the treated (ATT), which is the estimator of the net effect of tax avoidance on firm value. ATT is measured as follows:

$$
\begin{aligned}
ATT &= E[Y_{1i} - Y_{0i}|D_i = 1] \\
&= E\{E[Y_{1i} - Y_{0i}|D_i = 1, p(X_i)]\} \\
&= E\{E[Y_{1i}|D_i = 1, p(X_i)] - E[Y_{0i}|D_i = 0, p(X_i)]|D_i = 1\}
\end{aligned}
\tag{5}
$$

where $Y_{1i}$ and $Y_{0i}$ represent the outcome variable with tax avoidance and without tax avoidance, respectively.

Table 7 shows the ATTs based on different matching methods for both pre-matching and post-matching. After matching, most of the ATTs are negative and significant at the 5% level, indicating that tax avoidance indeed had a negative effect on firm value. Moreover, the t value and the significance of ATTs are reduced after matching, which reflects the existence of sample selection bias in our regression model before matching.

**Table 7.** Comparison of the average effects of treatment on the treated (ATTs).

| Variable | Matching Method | Sample | Tax-Avoidance Group | Control Group | ATT | t |
|---|---|---|---|---|---|---|
| TQ | Nearest-Neighbor Matching (NNM) | Pre-matching | 0.133 | 0.384 | −0.251 | −3.92 *** |
| | | Post-matching | 0.133 | 0.223 | −0.090 | −2.34 ** |
| | Radius Matching (RM) | Pre-matching | 0.126 | 0.335 | −0.209 | −2.22 ** |
| | | Post-matching | 0.126 | 0.207 | −0.081 | −1.67 * |
| | Kernel Matching (KM) | Pre-matching | 0.147 | 0.393 | −0.246 | −3.38 *** |
| | | Post-matching | 0.147 | 0.341 | −0.194 | −2.14 ** |

Note: ***, ** and * represent significance at the 1%, 5% and 10% level.

To examine Hypothesis 1, we further classified the tax-avoidance group into two subgroups. Firms in the top 75th percentile of CSR were defined as the high-level subgroup, and observations in the bottom 25th percentile were assigned to the low-level group. Table 8 shows the ATTs of the two groups in the post-matching analysis. We found that ATTs were positive and significant at the 5% level for the high-CSR group, whereas most ATTs are negative and non-significant at the 10% level for the low-CSR group. Hypothesis 1 is therefore generally supported.

**Table 8.** Comparison of high-CSR and low-CSR subgroups on ATTs.

| Variable | Matching Method | Full Sample | | High-CSR | | Low-CSR | |
|---|---|---|---|---|---|---|---|
| | | ATT | t | ATT | t | ATT | t |
| TQ | NNM | −0.090 | −2.34 ** | 0.078 | 2.49 ** | −0.116 | −1.84 * |
| | RM | −0.081 | −1.67 * | 0.041 | 1.96 ** | −0.096 | −1.18 |
| | KM | −0.194 | −2.14 ** | 0.036 | 2.01 *** | −0.174 | −1.79 * |

Note: ***, ** and * represent significance at the 1%, 5% and 10% level.

To examine Hypothesis 2, we classified the tax-avoidance group into a greenwashing subgroup and a non-greenwashing subgroup. Firms in the top 75th percentile of GW are assigned to the greenwashing subgroup. Otherwise, they are in the non-greenwashing subgroup. Table 9 shows the results of the ATTs of the two groups in the post-matching analysis. We found that all the ATTs were negative and significant at the 10% level for the greenwashing group, whereas most ATTs are positive and non-significant for the non-greenwashing group. Hypothesis 2 is generally supported.

**Table 9.** Comparison of greenwashing and non-greenwashing subgroups on ATTs.

| Variable | Matching Method | Full Sample | | Greenwashing | | Non-Greenwashing | |
|---|---|---|---|---|---|---|---|
| | | ATT | t | ATT | t | ATT | t |
| | NNM | −0.090 | −2.34 ** | −0.147 | −2.21 ** | 0.063 | 1.24 |
| TQ | RM | −0.081 | −1.67 * | −0.115 | −1.84 * | 0.087 | 1.32 |
| | KM | −0.194 | −2.14 ** | −0.138 | −2.45 ** | 0.045 | 1.98 * |

Note: ***, ** and * represent significance at the 1%, 5% and 10% level.

## 6. Discussion

As stated in the introduction, the central theme of the present research is to examine the risk management role of CSR engagement in promoting the value of tax avoidance. By comparing with previous studies that focused on CSR as a risk management tool against the risks of tax avoidance [2], we explored the insurance mechanism of CSR on firm value when firms engaged in tax avoidance. The present study not only investigated the insurance mechanism of CSR engagement, but also compared the effect of different CSR communication strategies, in particular by leveraging the concept of greenwashing. Our findings show that CSR engagement does have an insurance mechanism for firm value, and a greenwashing strategy also does not appear to produce such a mechanism.

Our findings are consistent with previous studies in the following respects. First, our findings show that tax avoidance leads to a decrease in firm value (see Table 4; Table 5). There is a positive effect for firms with high-level CSR engagement (firms that are more capable of managing reputation risk), while there is a negative effect for firms with low-level CSR engagement (firms that are less capable of managing reputation risk) (see Figure 1 and Table 6). The results are consistent with the hypothesis that tax avoidance can lead to decreases in financial performance by posing a significant risk to corporate reputation [3].

Second, our findings indicate that CSR engagement improves the value of tax avoidance by affecting investors' assessment of tax avoidance risks. It is consistent with the studies on the ex ante insurance mechanism of CSR engagement [13,45], which argued that investors evaluate CSR as a positive, risk-mitigating mechanism, even if no negative event actually takes place. In this way, CSR engagement enhances investors' positive assessments on tax avoidance.

Third, our findings show that, compared with an overaggressive communication strategy (greenwashing), a modest communication strategy (non-greenwashing, i.e., mere walk, walk the talk, and neither walk nor talk) is more effective at producing the insurance mechanism. This is consistent with previous studies, suggesting that it is important to avoid the impression of opportunism or ingratiation when adopting CSR, otherwise the insurance mechanism will be reduced [13,15].

These findings make a contribution to both the accounting and the management literature. First, this study contributes to the accounting literature by studying the value of tax avoidance. This study takes a risk management perspective, analyzing the impact of tax avoidance on firm value for firms with high or low CSR reputation. In this way, our research findings might imply a solution to the "under-sheltering puzzle": Why do some firms engage in tax avoidance, while others do not? [24,25]. That is to say, investors will doubt the value of tax avoidance in the absence of reputation management measures. These results suggest that firms lacking CSR engagement should forgo tax avoidance to avoid jeopardizing their good name.

Second, this study contributes to the risk management branch of CSR research by analyzing the ex ante insurance mechanism of CSR. Previous studies in this field have mainly taken an ex post perspective, examining the effectiveness of CSR in buffering against losses caused by negative events. It remains largely unexplored whether the insurance mechanism of CSR is valued by investors when no negative event has happened. Our findings show that investors indeed consider CSR to be an insurance mechanism and evaluate it positively before tax penalties and reputation losses have actually taken place. This is consistent with the studies on the ex ante insurance mechanism of CSR [13,46].

Third, this study contributes to the literature by discussing the conditions for CSR producing an insurance mechanism through a detailed investigation of greenwashing. A previous study has suggested that some characteristics of stakeholders, firms, or events can mitigate the insurance mechanism of CSR [15]. This study, from a CSR communication perspective, compares the insurance effectiveness between greenwashing and non-greenwashing strategies. We argue that a firm cannot benefit from greenwashing. This is because a greenwashing strategy might be interpreted by stakeholders as a signal conveying a corporate executive's intention of self-interest and opportunism. As a result, a greenwashing strategy cannot generate moral capital which is crucial to produce the insurance mechanism.

More generally, the results of this study contribute to resolving the debate on whether firms should devote limited resources to engaging in CSR [70,71]. Our findings imply that corporate executives need to be careful when launching substantive and symbolic actions. Substantive CSR actions may increase the cost but are effective at gaining moral legitimacy; a merely symbolic CSR action costs less and involves fewer resources, but may result in potential reputation loss, as it may make stakeholders suspect a firm's sincerity and actual CSR performance.

## 7. Conclusions

Overall, the results of this study suggest that the effect of tax avoidance on firm value varies depending on a firm's CSR engagement and CSR communication strategy (greenwashing vs. non-greenwashing). This study has documented the fact that tax avoidance is associated with reputation risks. By taking different CSR communication strategies into consideration, this paper further discussed the condition of CSR's insurance mechanism. We suggest that moral legitimacy is essential for CSR to produce an insurance mechanism, and a non-greenwashing strategy is more likely to gain moral legitimacy than a greenwashing strategy. This study may enrich the understanding of the connection between tax avoidance and financial performance.

Our research presents several limitations as follows: first, tax avoidance can also lead to tax authority penalties and agency costs, which impact firm value and CSR. Tax-avoidance firms carry a higher risk of tax authority punishment, which directly lead to cash outflow. Moreover, when firms engage in tax avoidance, executives must ensure that tax avoidance behaviors are obscured from tax authorities, which can afford them opportunities to pursue personal interests and lead investors to doubt the value of tax avoidance [72]. Future research should study the mediation role of tax authority penalty and agency cost between tax avoidance and financial performance.

Second, this study takes CSR as an exogenous variable. In light of tax authority penalties, tax-avoidance firms may leverage CSR strategies to reduce tax authority supervision. Chinese governments in different levels support firms engaging in CSR, and CSR is a part of the local government's performance appraisal. Thus, CSR firms are more likely to develop a closer relationship with the government, which can weaken tax authority enforcement [73,74]. To benefit from government support, firms may participate in CSR substantively or just communicate symbolically [61]. We suggest that future studies should consider the factors influencing CSR and greenwashing strategies.

Third, this study is based on a sample with a short period and small size. We suggest that future studies should extend the sample period and size and determine whether our findings are general for other years and firms.

**Author Contributions:** Conceptualization, W.L. (Wei Li), Y.L. and W.L. (Weining Li); Formal analysis, W.L. (Wei Li) and Y.L.; Writing—original draft, W.L. (Wei Li) and Y.L.; Writing—review & editing, Y.L. and W.L. (Weining Li).

**Funding:** National Natural Science Foundation of China (NSFC) 71772068; Humanity and Social Science Program Foundation of the Ministry of Education of China 17YJA630044.

**Conflicts of Interest:** The authors declare no conflict of interest.

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
