# Peer review of "Does CSR Action Provide Insurance-Like Protection to Tax-Avoiding Firms? Evidence from China"

_sustainability, doi:10.3390/su11195297_

Round 1
Reviewer 1 Report
The paper examines the effect of CSR on the relationship between tax avoidance and firm value. It shows that CSR mitigates the negative effect of tax avoidance on firm value while greenwashing CSR strategy exacerbates the negative effect of tax avoidance on firm value.
I really enjoyed reading the paper.
Firm value: the paper uses Tobin’s Q to measure firm value, the dependent variable. However, in many tax-avoidance studies, Tobin’s Q, reflects firms’ growth potential, is used as a controlled variable. The author(s) should justify this measurement.
Tax avoidance: CASH ETR should be used to measure tax avoidance since it measures how much cash tax saved. GAAP ETR is an accounting measure, which may be affected by some accounting treatments.
Reviewer 2 Report
Comments and suggestions for authors
This study examines whether firms engaging in tax avoidance can benefit from CSR.
I think the topic is very interesting and can help to open new lines of research. However, I would like to make the following observations so that you can consider them in order to improve your paper:
Line 31, we can see “However, recent studies have noted that tax avoidance is usually regarded as socially irresponsible and unethical by the public and the media”. The articles are from 2009, 2013 and 2014. Are there recently articles on this subject?
Line 39, we can see: “CSR has been referred to as a firm’s voluntary contribution to the development of a better society though various activities, such as philanthropy, environmental and/or charity projects, and the improvement of social welfare, beyond business operations [10,11]”. I think that social action is confused with corporate social responsibility. Explain better what CSR is.
Line 95: “Previous studies have suggested that, in some conditions, CSR action may lead to negative evaluations from stakeholders, and pointed out characteristics of stakeholders, firms, or events as the main conditions”. There are no references to such studies. Why? What are these studies?
Methodology should be shortly mentioned in the Abstract.
It has been unclear to me that there is no specific section called methodology, explaining why a regression has been carried out (and not a structural equation model for example) and which statistical programme has been used.
Line 369: “Observations in the top 75th percentile of CSR and GW are defined as high-level groups, and observations in the bottom 25th percentile of CSR and GW are low-level groups”. Why? In line 275 we can see: “The higher the GW, the more likely it is that a firm adopts a greenwashing strategy; the lower the GW, the more likely it is that a firm adopts a nongreenwashing strategy”. This is contradictory.
Rethink the Discussion and Conclusions. At the moment it lacks the limitations and the future research: define the limitations of your research and propose possible new or expanded ways of thinking about your research problem.
The conclusions section is the most important part of a paper and in this case it is too short.
I do hope you find the comments helpful as you move forward with your paper.
Reviewer 3 Report
There are several papers already that examine the relation between tax avoidance and CSR and/or firm value. You need to differentiate your study from that of previous work.
The paper could be improved if:
You do not introduce the risk management perspective of CSR-this is already well established in the literature.Your contribution lies more with the concepts greenwashing and insurance effects of CSR.
Include your model in the paper. Why the sample period 2013-2016?
Further tests are required: It would be nice to conduct subgroup analysis based on high vs low tax avoidance, propensity score matching.
My additional comments are:
1) why is the study restricted to the period 2013-2016?
2) Why focus on China?
3) The risk management perspective of CSR use is already well established in the literature. This should be drawn out more.
4) The study argues that tax avoidance gives rise to reputation risk. There are also other costs and benefits which may impact on firms strategy to greenwash or not and on firm value. These should be discussed.
5) Include the regression model in the paper
Round 2
Reviewer 1 Report
The authors addressed the issues and the manuscript has been improved.
Reviewer 2 Report
Dear Authors,
With the changes introduced, your article has been greatly improved. Congratulations!
Best regards
Reviewer 3 Report
The authors have made the paper much better. They have addressed my comments adequately.